# Principles of Riemannian Geometry in Neural Networks

**Michael Hauser**
Department of Mechanical Engineering
Pennsylvania State University
State College, PA 16801
mzh190@psu.edu

**Asok Ray**
Department of Mechanical Engineering
Pennsylvania State University
State College, PA 16801
axr2@psu.edu

## Abstract

This study deals with neural networks in the sense of geometric transformations acting on the coordinate representation of the underlying data manifold which the data is sampled from. It forms part of an attempt to construct a formalized general theory of neural networks in the setting of Riemannian geometry. From this perspective, the following theoretical results are developed and proven for feedforward networks. First it is shown that residual neural networks are finite difference approximations to dynamical systems of first order differential equations, as opposed to ordinary networks that are static. This implies that the network is learning systems of differential equations governing the coordinate transformations that represent the data. Second it is shown that a closed form solution of the metric tensor on the underlying data manifold can be found by backpropagating the coordinate representations learned by the neural network itself. This is formulated in a formal abstract sense as a sequence of Lie group actions on the metric fibre space in the principal and associated bundles on the data manifold. Toy experiments were run to confirm parts of the proposed theory, as well as to provide intuitions as to how neural networks operate on data.

## 1  Introduction

The introduction is divided into two parts. Section 1.1 attempts to succinctly describe ways in which neural networks are usually understood to operate. Section 1.2 articulates a more minority perspective. It is this minority perspective that this study develops, showing that there exists a rich connection between neural networks and Riemannian geometry.

### 1.1  Latent variable perspectives

Neural networks are usually understood from a latent variable perspective, in the sense that successive layers are learning successive representations of the data. For example, convolution networks [10] are understood quite well as learning hierarchical representations of images [19]. Long short-term memory networks [9] are designed such that input data act on a memory cell to avoid problems with long term dependencies. More complex devices like neural Turing machines are designed with similar intuitions for reading and writing to a memory [6].

Residual networks were designed [7] with the intuition that it is easier to learn perturbations from the identity map than it is to learn an unreferenced map. Further experiments then suggest that residual networks work well because, during forward propagation and back propagation, the signal from any block can be mapped to any other block [8]. After unraveling the residual network, this attribute can be seen more clearly. From this perspective, the residual network can be understood as an ensemble of shallower networks [17].

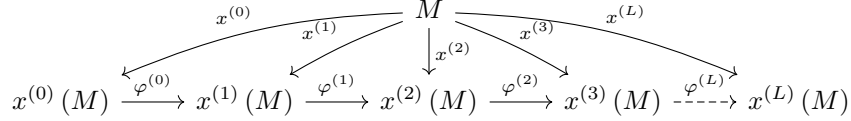

Figure 1: Coordinate systems $x^{(l+1)} := \varphi^{(l)} \circ ... \circ \varphi^{(1)} \circ \varphi^{(0)} \circ x^{(0)}$ induced by the coordinate transformations $\varphi^{(l)} : x^{(l)}(M) \to \left(\varphi^{(l)} \circ x^{(l)}\right)(M)$ learned by the neural network. The pullback metric $g_{x^{(l)}(M)}(X,Y) := g_{\left(\varphi^{(l)} \circ x^{(l)}\right)(M)}\left(\varphi^{(l)}_* X, \varphi^{(l)}_* Y\right)$ pulls-back (i.e. backpropagates) the coordinate representation of the metric tensor from layer $l+1$ to layer $l$, via the pushforward map $\varphi^{(l)}_* : Tx^{(l)}(M) \to T\left(\varphi^{(l)} \circ x^{(l)}\right)(M)$ between tangent spaces.

## 1.2 Geometric perspectives

These latent variable perspectives are a powerful tool for understanding and designing neural networks. However, they often overlook the fundamental process taking place, where successive layers successively warp the coordinate representation of the data manifold with nonlinear transformations into a form where the classes in the data manifold are linearly separable by hyperplanes. These nested compositions of affine transformations followed by nonlinear activations can be seen by work done by C. Olah (http://colah.github.io/) and published by LeCun et al. [11].

Research in language modeling has shown that the word embeddings learned by the network preserve vector offsets[13], with an example given as $x_{apples} - x_{apple} \approx x_{cars} - x_{car}$ for the word embedding vector $x_i$. This suggests the network is learning a word embedding space with some resemblance to group closure, with group operation vector addition. Note that closure is generally not a property of data, for if instead of word embeddings one had images of apples and cars, preservation of these vector offsets would certainly not hold at the input [3]. This is because the input images are represented in Cartesian coordinates, but are not sampled from a flat data manifold, and so one should not measure vector offsets by Euclidean distance. In Locally Linear Embedding [14], a coordinate system is learned in which Euclidean distance can be used. This work shows that neural networks are also learning a coordinate system in which the data manifold can be measured by Euclidean distance, and the coordinate representation of the metric tensor can be backpropagated through to the input so that distance can be measured in the input coordinates.

## 2 Mathematical notations

Einstein notation is used throughout this paper. A raised index in parenthesis, such as $x^{(l)}$, means it is the $l^{th}$ coordinate system while $\varphi^{(l)}$ means it is the $l^{th}$ coordinate transformation. If the index is not in parenthesis, a superscript free index means it is components of a vector, a subscript free index means it is components of a covector, and a repeated index means implied summation. The . in tensors, such as $A^{a\cdot}_{\cdot b}$, are placeholders to keep track of which index comes first, second, etc.

A (topological) manifold $M$ of dimension $\dim M$ is a Hausdorff, paracompact topological space that is locally homeomorphic to $\mathbb{R}^{\dim M}$ [18]. This homeomorphism $x : U \to x(U) \subseteq \mathbb{R}^{\dim M}$ is called a coordinate system on $U \subseteq M$. Non-Euclidean manifolds, such as $S^1$, can be created by taking an image and rotating it in a circle. A feedforward network learns coordinate transformations $\varphi^{(l)} : x^{(l)}(M) \to \left(\varphi^{(l)} \circ x^{(l)}\right)(M)$, where the new coordinates $x^{(l+1)} := \varphi^{(l)}\left(x^{(l)}\right) : M \to x^{(l+1)}(M)$, and is initialized in Cartesian coordinates $x^{(0)} : M \to x^{(0)}(M)$, as seen in Figure 1.

A data point $q \in M$ can only be represented as numbers with respect to some coordinate system; with the coordinates at layer $l+1$, $q$ is represented as the layerwise composition $x^{(l+1)}(q) := \left(\varphi^{(l)} \circ ... \circ \varphi^{(1)} \circ \varphi^{(0)} \circ x^{(0)}\right)(q)$. The output coordinate representation is $x^{(L)}(M) \subseteq \mathbb{R}^d$.

For an activation function $f$, such as ReLU or $\tanh$, a standard feedforward network transforms coordinates as $x^{(l+1)} := \varphi^{(l)}\left(x^{(l)}\right) := f(x^{(l)}; l)$. Note ReLu is not a bijection and thus not a proper coordinate transformation. A residual network transforms coordinates as $x^{(l+1)} := \varphi^{(l)}\left(x^{(l)}\right) := x^{(l)} + f(x^{(l)}; l)$. Note that these are global coordinates over the entire manifold. A residual network with ReLu activation is bijective, and is piecewise linear with kinks of infinite curvature.

With the Softmax coordinate transformation defined as $\text{softmax}\left(W^{(L)} \cdot x^{(L)}\right)^j := e^{W^{(L)j}x^{(L)}} / \sum_{k=1}^{K} e^{W^{(L)k}x^{(L)}}$ the probability of $q \in M$ being from class $j$ is $P\left(Y = j | X = q\right) = \text{softmax}\left(W^{(L)} \cdot x^{(L)}(q)\right)^j$.

## 3  Neural networks as $\mathcal{C}^k$ differentiable coordinate transformations

One can define entire classes of coordinate transformations. The following formulation also has the form of differentiable curves/trajectories, but because the number of dimensions often changes as one moves through the network, it is difficult to interpret a trajectory traveling through a space of changing dimensions. A standard feedforward neural network is a $\mathcal{C}^0$ function:

$$x^{(l+1)} := f(x^{(l)}; l) \tag{1}$$

A residual network has the form $x^{(l+1)} = x^{(l)} + f(x^{(l)}; l)$. However, because of eventually taking the limit as $L \to \infty$ and $l \in [0, 1] \subset \mathbb{R}$, as opposed to $l$ being only a finitely countable index, the equivalent form of the residual network is as follows:

$$x^{(l+1)} \simeq x^{(l)} + f(x^{(l)}; l)\Delta l \tag{2}$$

where $\Delta l = 1/L$ for a uniform partition of the interval $[0, 1]$ and is implicit in the weight matrix.

One can define entire classes of coordinate transformations inspired by finite difference approximations of differential equations. These can be used to impose $k^{th}$ order differentiable smoothness:

$$\delta x^{(l)} := x^{(l+1)} - x^{(l)} \simeq f(x^{(l)}; l)\Delta l \tag{3}$$

$$\delta^2 x^{(l)} := x^{(l+1)} - 2x^{(l)} + x^{(l-1)} \simeq f(x^{(l)}; l)\Delta l^2 \tag{4}$$

Each of these define a differential equation, but of different orders of smoothness on the coordinate transformations. Written in this form the residual network in Equation 3 is a first-order forward difference approximation to a $\mathcal{C}^1$ coordinate transformation and has $\mathcal{O}\left(\Delta l\right)$ error. Network architectures with higher order accuracies can be constructed, such as central differencing approximations of a $\mathcal{C}^1$ coordinate transformation to give $\mathcal{O}\left(\Delta l^2\right)$ error.

Note that the architecture of a standard feedforward neural network is a static equation, while the others are dynamic. Also note that Equation 4 can be rewritten $x^{(l+1)} = x^{(l)} + f(x^{(l)}; l)\Delta l^2 + \delta x^{(l-1)}$, where $\delta x^{(l-1)} = x^{(l)} - x^{(l-1)}$, and in this form one sees that this is a residual network with an extra term $\delta x^{(l-1)}$ acting as a sort of momentum term on the coordinate transformations. This momentum term is explored in Section 7.1.

By the definitions of the $\mathcal{C}^k$ networks given by Equations 3-4, the right hand side is both continuous and independent of $\Delta l$ (after dividing), and so the limit exists as $\Delta l \to 0$. Convergence rates and error bounds of finite difference approximations can be applied to these equations. By the standard definition of the derivative, the residual network defines a system of differentiable transformations.

$$\frac{dx^{(l)}}{dl} := \lim_{\Delta l \to 0} \frac{x^{(l+\Delta l)} - x^{(l)}}{\Delta l} = f(x^{(l)}; l) \tag{5}$$

$$\frac{d^2 x^{(l)}}{dl^2} := \lim_{\Delta l \to 0} \frac{x^{(l+\Delta l)} - 2x^{(l)} + x^{(l-\Delta l)}}{\Delta l^2} = f(x^{(l)}; l) \tag{6}$$

Notations are slightly changed, by taking $l = n\Delta l$ for $n \in \{0, 1, 2, .., L - 1\}$ and indexing the layers by the fractional index $l$ instead of the integer index $n$. This defines a partitioning:

$$\mathcal{P} = \{0 = l(0) < l(1) < l(2) < ... < l(n) < ... < l(L) = 1\} \tag{7}$$

where $\Delta l(n) := l(n + 1) - l(n)$ can in general vary with $n$ as the $\max_n \Delta l(n)$ still goes to zero as $L \to \infty$. To reduce notation, this paper will write $\Delta l := \Delta l(n)$ for all $n \in \{0, 1, 2, ..., L - 1\}$.

In [4], a deep residual convolution network was trained on ImageNet in the usual fashion except parameter weights between residual blocks at the same dimension were shared, at a cost to the accuracy of only $0.2\%$. This is the difference between learning an inhomogeneous first order equation $\frac{dx^{(l)}}{dl} := f(x^{(l)}; l)$ and a (piecewise) homogeneous first order equation $\frac{dx^{(l)}}{dl} := f(x^{(l)})$.

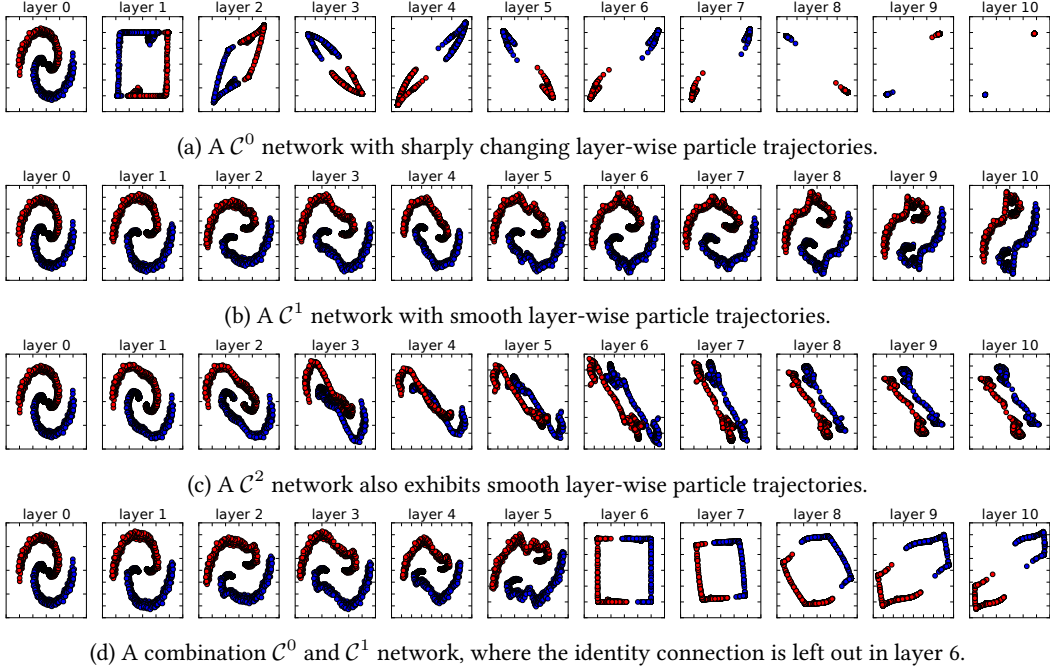

(a) A $\mathcal{C}^0$ network with sharply changing layer-wise particle trajectories.

(b) A $\mathcal{C}^1$ network with smooth layer-wise particle trajectories.

(c) A $\mathcal{C}^2$ network also exhibits smooth layer-wise particle trajectories.

(d) A combination $\mathcal{C}^0$ and $\mathcal{C}^1$ network, where the identity connection is left out in layer 6.

Figure 2: Untangling the same spiral with 2-dimensional neural networks with different constraints on smoothness. The $x$ and $y$ axes are the two nodes of the neural network at a given layer $l$, where layer 0 is the input data. The $\mathcal{C}^0$ network is a standard network, while the $\mathcal{C}^1$ network is a residual network and the $\mathcal{C}^2$ network also exhibits smooth layerwise transformations. All networks achieve 0.0% error rates. The momentum term in the $\mathcal{C}^2$ network allows the red and blue sets to pass over each other in layers 3, 4 and 5. Figure 2d has the identity connection for all layers other than layer 6.

## 4 The Riemannian metric tensor learned by neural networks

From the perspective of differentiable geometry, as one moves through the layers of the neural network, the data manifold stays the same but the coordinate representation of the data manifold changes with each successive affine transformation and nonlinear activation. The objective of the neural network is to find a coordinate representation of the data manifold such that the classes are linearly separable by hyperplanes.

**Definition 4.1.** (Riemannian manifold [18]) A Riemannian manifold $(M, g)$ is a real smooth manifold $M$ with an inner product, defined by the positive definite metric tensor $g$, varying smoothly on the tangent space of $M$.

If the network has been well trained as a classifier, then by Euclidean distance two input points of the same class may be far apart when represented by the input coordinates but close together in the output coordinates. Similarly, two points of different classes may be near each other when represented by the input coordinates but far apart in the output coordinates. These ideas form the basis of Locally Linear Embeddings [14]. The intuitive way to measure distances is in the output coordinates, which even in the unsupervised case tends to be a flattened representation of the data manifold [3]. Accordingly, the metric in the output coordinates is the Euclidean metric:

$$g\left(x^{(L)}\right)_{a_L b_L} := \eta_{a_L b_L} \tag{8}$$

The elements of the metric tensor transforms as a tensor with coordinate transformations:

$$g(x^{(l)})_{a_l b_l} = \left(\frac{\partial x^{(l+1)}}{\partial x^{(l)}}\right)^{a_{l+1}\cdot}_{\cdot a_l} \left(\frac{\partial x^{(l+1)}}{\partial x^{(l)}}\right)^{b_{l+1}\cdot}_{\cdot b_l} g(x^{(l+1)})_{a_{l+1} b_{l+1}} \tag{9}$$

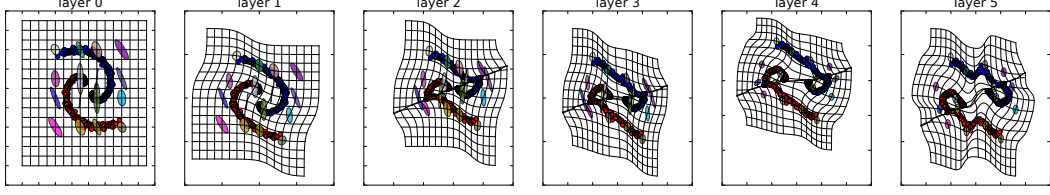

Figure 3: A $\mathcal{C}^1$ (residual) network with a hyperbolic tangent activation function separating the spiral manifold. Additionally, balls of constant radius $ds = \sqrt{g_{a_l b_l}(x^{(l)})dx^{(l)a_l}dx^{(l)b_l}}$ at different points are shown. In the output coordinates, distances are measured by the standard Euclidean metric in Equation 8, and so the circles are "round". The coordinate representation of the metric tensor is pulled-back (backpropagated) through the network to the input by Equations 9 and 10. Distances on the data manifold can then be measured with the input Cartesian coordinates, and so the circles are not round. These balls can also be interpreted as forming an $\epsilon - \delta$ relationship across layers of the network, where an $\epsilon$-ball at one layer corresponds to a $\delta$-ball at the previous layer.

The above recursive formula is solved from the output layer to the input, i.e. the coordinate representation of the metric tensor is backpropagated through the network from output to input:

$$g(x^{(l)})_{a_l b_l} = \prod_{l'=L-1}^{l} \left[ \left(\frac{\partial x^{(l'+1)}}{\partial x^{(l')}}\right)^{a_{l'+1}\cdot}_{\quad .a_{l'}} \left(\frac{\partial x^{(l'+1)}}{\partial x^{(l')}}\right)^{b_{l'+1}\cdot}_{\quad .b_{l'}} \right] \eta_{a_L b_L} \tag{10}$$

If the network is taken to be residual as in Equation 2, then the Jacobian of the coordinate transformation is found, with $\delta^{a_{l+1}\cdot}_{.a_l}$ the Kronecker delta:

$$\left(\frac{\partial x^{(l+1)}}{\partial x^{(l)}}\right)^{a_{l+1}\cdot}_{\quad .a_l} = \delta^{a_{l+1}\cdot}_{.a_l} + \left(\frac{\partial f\left(x^{(l)};l\right)}{\partial x^{(l)}}\right)^{a_{l+1}\cdot}_{\quad .a_l} \Delta l \tag{11}$$

Backpropagating the coordinate representation of the metric tensor requires the sequence of matrix products from output to input, and can be defined for any layer $l$:

$$P^{a_L \cdot}_{.a_l} := \prod_{l'=l}^{L-1} \left[ \delta^{a_{l'+1}\cdot}_{.a_{l'}} + \left(\frac{\partial f(z^{(l'+1)};l')}{\partial z^{(l'+1)}}\right)^{a_{l'+1}\cdot}_{\quad .e_{l'+1}} \left(\frac{\partial z^{(l'+1)}}{\partial x^{(l')}}\right)^{e_{l'+1}\cdot}_{\quad .a_{l'}} \Delta l \right] \tag{12}$$

where $z^{(l+1)} := W^{(l)} \cdot x^{(l)} + b^{(l)}$. With this, taking the output metric to be the standard Euclidean metric $\eta_{ab}$, the linear element can be represented in the coordinate space for any layer $l$:

$$ds^2 = \eta_{ab} P^{a\cdot}_{.a_l} P^{b\cdot}_{.b_l} dx^{a_l} dx^{b_l} \tag{13}$$

The data manifold is independent of coordinate representation. At the output where distances are measured by the standard Euclidean metric an $\epsilon$-ball can be defined. The linear element in Equation 13 defines the corresponding $\delta$-ball at layer $l$. This can be used to see what in the input space the neural network encodes as similar in the output space.

As $L \to \infty$, Equation 12 becomes an infinite product of matricies (from our infinite applications of the chain rule) and these transformations act smoothly along the fibres of the tensor bundle. The proof that this sequence converges in the limit can be found in the appendix.

This analysis has so far assumed a constant layerwise dimension, which is not how most neural networks are used in practice, where the number of nodes often changes. This is handled by the pullback metric [18]. Manifolds can be submersed and immersed into lower and higher dimensional spaces so long as the rank of the pushforward Jacobian matrix is constant for every $p \in M$ [12]. The dimension of the underlying data manifold is defined as the dimension of the smallest, bottleneck layer of the neural network, i.e. $\dim M := \min_l \dim x^{(l)}(M)$, and all other higher dimensional layers are immersion/embedding representations of this lowest dimensional representation.

**Definition 4.2.** (Pushforward map) Let $M$ and $N$ be topological manifolds, $\varphi^{(l)} : M \to N$ a smooth map and $TM$ and $TN$ be their respective tangent spaces. Also let $X \in TM$ where $X : \mathcal{C}^\infty(M) \to \mathbb{R}$, and $f \in \mathcal{C}^\infty(N)$. The pushforward is the linear map $\varphi_*^{(l)} : TM \to TN$ that takes an element $X \mapsto \varphi_*^{(l)} X$ and is defined by its action on $f$ as $\left(\varphi_*^{(l)} X\right)(f) := X\left(f \circ \varphi^{(l)}\right)$.

**Definition 4.3.** (Pullback metric) Let $(M, g_M)$ and $(N, g_N)$ be Riemannian manifolds, $\varphi^{(l)} : M \to N$ a smooth map and $\varphi_*^{(l)} : TM \to TN$ the pushforward between their tangent spaces $TM$ and $TN$. Then the pullback metric on $M$ is given by $g_M(X, Y) := g_N\left(\varphi_*^{(l)} X, \varphi_*^{(l)} Y\right) \forall X, Y \in TM$.

In practice being able to change dimensions in the neural network is important for many reasons. One reason is that neural networks usually have access to a limited number of types of nonlinear coordinate transformations, for example $\tanh$, $\sigma$ and ReLU. This severely limits the ability of the network to separate the wide variety of manifolds that exist. For example, the networks have difficulty linearly separating the simple toy spirals in Figures 2 because they only have access to coordinate transformations of the form $\tanh$. If instead they had access to a coordinate system that was more appropriate for spirals, such as polar coordinates, they could very easily separate the data. This is the reason why Locally Linear Embeddings [14] could very easily discover the coordinate charts for the underlying manifold, because k-nearest neighbors is an extremely flexible type of nonlinearity. Allowing the network to go into higher dimensions makes it easier to separate data.

# 5 Lie Group actions on the metric fibre bundle

This section will abstractly formulate Section 4 as neural networks learning sequences of left Lie Group actions on the metric (fibre) space over the data manifold to make the metric representation of the underlying data manifold Euclidean. Several definitions, which can be found in the appendix in the full version of this paper, are needed to formulate Lie group actions on principal and associated fibre bundles, namely of bundles, fibre bundles, Lie Groups and their actions on manifolds [18].

**Definition 5.1.** (Principal fibre bundle) A bundle $(E, \pi, M)$ is called a principal $G$-bundle if:

(i.) $E$ is equipped with a right $G$-action $\lhd: E \times G \to E$.

(ii.) The right $G$-action $\lhd$ is free.

(iii.) $(E, \pi, M)$ is (bundle) isomorphic to $(E, \rho, E/G)$ where the surjective projection map $\rho : E \to E/G$ is defined by $\rho(\epsilon) := [\epsilon]$ as the equivalence class of points of $\epsilon$

*Remark.* (Principal bundle) The principal fibre bundle can be thought of (locally) as a fibre bundle with fibres $G$ over the base manifold $M$.

**Definition 5.2.** (Associated fibre bundle) Given a $G$ principal bundle and a smooth manifold $F$ on which exists a left $G$-action $\rhd: G \times F \to F$, the associated fibre bundle $(P_F, \pi_F, M)$ is defined as follows:

(i.) let $\sim_G$ be the relation on $P \times F$ defined as follows:

$(p, f) \sim_G (p', f') : \iff \exists h \in G : p' = p \lhd h$ and $f' = h^{-1} \rhd f$, and thus $P_F := (P \times F) / \sim_G$.

(ii.) define $\pi_F : P_F \to M$ by $\pi_F([(p, f)]) := \pi(p)$

Neural network actions on the manifold $M$ are a (layerwise) sequence of left $G$-actions on the associated (metric space) fibre bundle. Let the dimension of the manifold $d := \dim M$.

The structure group $G$ is taken to be the general linear group of dimension $d$ over $\mathbb{R}$, i.e.
$G = GL(d, \mathbb{R}) := \{\phi : \mathbb{R}^d \to \mathbb{R}^d | \det \phi \neq 0\}$.

The principal bundle $P$ is taken to be the frame bundle, i.e. $P = LM := \cup_{p \in M} L_p M := \cup_{p \in M}\{(e_1, ..., e_d) \in T_p M | (e_1, ..., e_d)$ is a basis of $T_p M\}$, where $T_p M$ is the tangent space of $M$ at the point $p \in M$.

The right $G$-action $\lhd: LM \times GL(d, \mathbb{R}) \to LM$ is defined by $e \lhd h = (e_1, ..., e_d) \lhd h := (h_{.1}^{a_l \cdot} e_{a_l}, ..., h_{.d}^{a_l \cdot} e_{a_l})$, which is the standard transformation law of linear algebra.

The fibre $F$ in the associated bundle will be the metric tensor space, and so $F = \left(\mathbb{R}^d\right)^* \times \left(\mathbb{R}^d\right)^*$, where the $*$ denotes the cospace. With this, the left $G$-action $\rhd: GL(d, \mathbb{R}) \times F \to F$ is defined as the inverse of the left, namely $\left(h^{-1} \rhd g\right)_{a_l b_l} := (g \lhd h)_{a_l b_l} = g_{a_{l+1} b_{l+1}} h_{.a_l}^{a_{l+1} \cdot} h_{.b_l}^{b_{l+1} \cdot}$.

Layerwise sequential applications of the left $G$-action from output to input is thus simply understood:

$$\left(h_0^{-1} \triangleright h_1^{-1} \triangleright ... \triangleright h_L^{-1} \triangleright g\right)_{a_0 b_0} = \left(h_0^{-1} \bullet ... \bullet h_L^{-1}\right) \triangleright g_{a_L b_L} = \prod_{l'=L-1}^{0} \left(h_{.a_l'}^{a_{l'+1}\cdot} h_{.b_{l'}}^{b_{l'+1}\cdot}\right) g_{a_L b_L}$$

(14)

This is equivalent to Equation 10, only formulated in a formal, abstract sense.

## 6  Backpropagation as a sequence of right Lie Group actions

A similar analysis that has been performed in Sections 4 and 5 can be done to generalize error backpropagation as a sequence of right Lie Group actions on the output error (or more generally pull-back the frame bundle). The discrete layerwise error backpropagation algorithm [15] is derived using the chain rule on graphs. The closed form solution of the gradient of the output error $E$ with respect to any layer weight $W^{(l-1)}$ can be solved for recursively from the output, by backpropagating errors:

$$\frac{\partial E}{\partial W^{(l-1)}} = \left(\frac{\partial E}{\partial x^{(L)}}\right)_{a_L} \prod_{l'=L-1}^{l} \left(\frac{\partial x^{(l'+1)}}{\partial x^{(l')}}\right)_{.a_{l'}}^{a_{l'+1}\cdot} \left(\frac{\partial x^{(l)}}{\partial W^{(l-1)}}\right)^{a_l}$$

(15)

In practice, one further applies the chain rule $\left(\frac{\partial x^{(l)}}{\partial W^{(l-1)}}\right)^{a_l} = \left(\frac{\partial x^{(l)}}{\partial z^{(l)}}\right)_{.b_l}^{a_l\cdot} \left(\frac{\partial z^{(l)}}{\partial W^{(l-1)}}\right)^{b_l}$. Note that $W^{(l-1)}$ is a coordinate chart on the parameter manifold [1], *not* the data manifold. In this form it is immediately seen that error backpropagation is a sequence of right $G$-actions $\prod_{l'=L-1}^{l} \left(\frac{\partial x^{(l'+1)}}{\partial x^{(l')}}\right)_{.a_{l'}}^{a_{l'+1}\cdot}$ on the output frame bundle $\left(\frac{\partial}{\partial x^{(L)}}\right)_{a_L}$. This pulls-back the frame bundle acting on $E$ to the coordinate system at layer $l$, and thus puts it in the same space as $\left(\frac{\partial x^{(l)}}{\partial W^{(l-1)}}\right)^{a_l}$.

For the residual network, the transformation matrix Equation 11 can be inserted into Equation 15. By the same logic as before, the infinite tensor product in Equation 15 converges in the limit $L \to \infty$ in the same way as in Equation 12, and so it is not rewritten here. In the limit this becomes a smooth right $G$-action on the frame bundle, which itself is acting on the error cost function.

## 7  Numerical experiments

This section presents the results of numerical experiments used to understand the proposed theory. The $\mathcal{C}^\infty$ hyperbolic tangent has been used for all experiments, with weights initialized according to [5]. For all of the experiments, layer 0 is the input Cartesian coordinate representation of the data manifold, and the final layer $L$ is the last hidden layer before the linear softmax classifier. GPU implementations of the neural networks are written in the Python library Theano [2, 16].

### 7.1  Neural networks with $\mathcal{C}^k$ differentiable coordinate transformations

As described in Section 3, $k^{th}$ order smoothness can be imposed on the network by considering network structures defined by e.g. Equations 3-4. As seen in Figure 2a, the standard $\mathcal{C}^0$ network with no impositions on differentiability has very sharp layerwise transformations and separates the data in an unintuitive way. The $\mathcal{C}^1$ residual network and $\mathcal{C}^2$ network can be seen in Figures 2b and 2c, and exhibit smooth layerwise transformations and separate the data in a more intuitive way. Forward differencing is used for the $\mathcal{C}^1$ network, while central differencing was used for the $\mathcal{C}^2$ network, except at the output layer where backward differencing was used, and at the input first order smoothness was used as forward differencing violates causality.

In Figure 2c one can see that for the $\mathcal{C}^2$ network the red and blue data sets pass over each other in layers 4, 5 and 6. This can be understood as the $\mathcal{C}^2$ network has the same form as a residual network, with an additional momentum term pushing the data past each other.

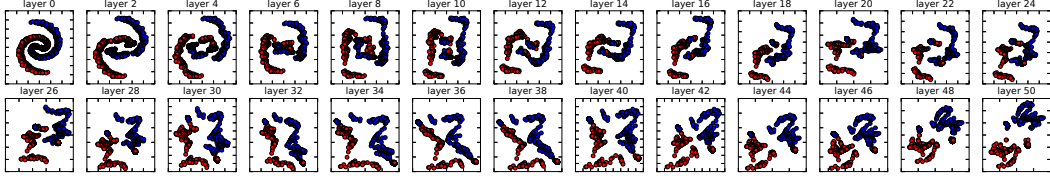

(a) A batch size of 300 for untangling data. As early as layer 4 the input connected sets have been disconnected and the data are untangled in an unintuitive way. This means a more complex coordinate representation of the data manifold was learned.

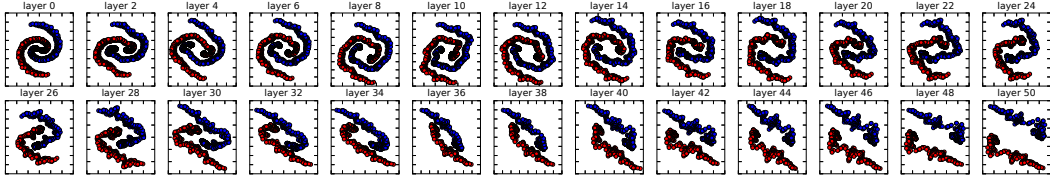

(b) A batch size of 1000 for untangling data. Because the large batch size can well-sample the data manifold, the spiral sets stay connected and are untangled in an intuitive way. This means a simple coordinate representation of the data manifold was learned.

Figure 4: The effect of batch size on coordinate representation learned by the same 2-dimensional $\mathcal{C}^1$ network, where layer 0 is the input representation, and both examples achieve $0\%$ error. A basic theorem in topology says continuous functions map connected sets to connected sets. A small batch size of 300 during training sparsely samples from the connected manifold and the network learns overfitted coordinate representations. With a larger batch size of 1000 during training the network learns a simpler coordinate representation and keeps the connected input connected throughout.

## 7.2 Coordinate representations of the data manifold and metric tensor

As described in Sections 4 and 5, the network is learning a sequence of non-linear coordinate transformations, beginning with Cartesian coordinates, to find a coordinate representation of the data manifold that well represents the data, and this representation tends to be flat. This process can be visualized in Figure 3. This experiment used a $\mathcal{C}^1$ (residual) network and so the group actions on the principal and associated bundles act approximately smoothly along the fibres of the bundles.

In the forward direction, beginning with Cartesian coordinates, a sequence of $\mathcal{C}^1$ differential coordinate transformations is applied to find a nonlinear coordinate representation of the data manifold such that in the output coordinates the classes satisfy the cost restraint. In the reverse direction, starting with a standard Euclidean metric at the output, Equation 8, the coordinate representation of the metric tensor is backpropagated through the network to the input by Equations 9-10 to find the metric tensor representation in the input Cartesian coordinates. The principal components of the metric tensor are used to draw the ellipses in Figure 3.

## 7.3 Effect of batch size on set connectedness and topology

A basic theorem in topology says that continuous functions map connected sets to connected sets. However, in Figure 4a it is seen that as early as layer 4 the continuous neural network is breaking the connected input set into disconnected sets. Additionally, and although it achieves $0\%$ error, it is learning very complicated and unintuitive coordinate transformations to represent the data in a linearly separable form. This is because during training with a small batch size of 300 in the stochastic gradient descent search, the underlying manifold was not sufficiently sampled to represent the entire connected manifold and so it seemed disconnected.

This is compared to Figure 4b in which a larger batch size of 1000 was used and was sufficiently sampled to represent the entire connected manifold, and the network was also able to achieve $0\%$ error. The coordinate transformations learned by the neural network with the larger batch size seem to more intuitively untangle the data in a simpler way than that of Figure 4a. Note that this experiment is in 2-dimensions, and with higher dimensional data the issue of batch size and set connectedness becomes exponentially more important by the curse of dimensionality.

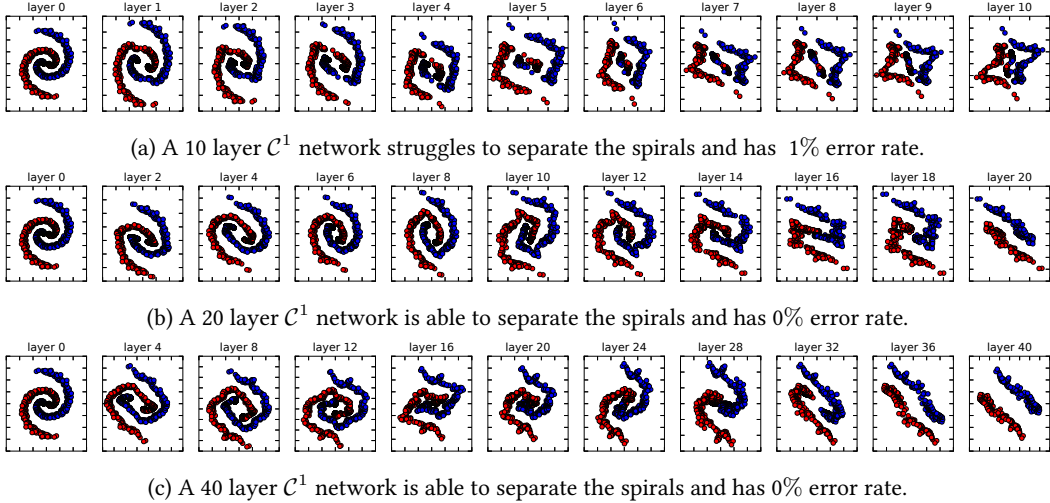

(a) A 10 layer $\mathcal{C}^1$ network struggles to separate the spirals and has $1\%$ error rate.

(b) A 20 layer $\mathcal{C}^1$ network is able to separate the spirals and has $0\%$ error rate.

(c) A 40 layer $\mathcal{C}^1$ network is able to separate the spirals and has $0\%$ error rate.

Figure 5: The effect of number of layers on the separation process of a $\mathcal{C}^1$ neural network. In Figure 5a it is seen that the $\Delta l$ is too large to properly separate the data. In Figures 5b and 5c the $\Delta l$ is sufficiently small to separate the data. Interestingly, the separation process is not as simple as merely doubling the parameterization and halving the partitioning in Equation 7 because this is a nonlinear system of ODE's. This is seen in Figures 5b and 5c; the data are at different levels of separation at the same position of layer parameterization, for example by comparing layer 18 in Figure 5b to layer 36 in Figure 5c.

## 7.4 Effect of number of layers on the separation process

This experiment compares the process in which 2-dimensional $\mathcal{C}^1$ networks with 10, 20 and 40 layers separate the same data, thus experimenting on the $\Delta l$ in the partitioning of Equation 7, as seen in Figure 5. The 10 layer network is unable to properly separate the data and achieves a $1\%$ error rate, whereas the 20 and 40 layer networks both achieve $0\%$ error rates. In Figures 5b and 5c it is seen that at same positions of layer parameterization, for example layers 18 and 36 respectively, the data are at different levels of separation. This implies that the partitioning cannot be interpreted as simply as halving the $\Delta l$ when doubling the number of layers. This is because the system of ODE's are nonlinear and the $\Delta l$ is implicit in the weight matrix.

## 8 Conclusions

This paper forms part of an attempt to construct a formalized general theory of neural networks as a branch of Riemannian geometry. In the forward direction, and starting in Cartesian coordinates, the network is learning a sequence of coordinate transformations to find a coordinate representation of the data manifold that well encodes the data, and experimental results suggest this imposes a flatness constraint on the metric tensor in this learned coordinate system. One can then backpropagate the coordinate representation of the metric tensor to find its form in Cartesian coordinates. This can be used to define an $\epsilon - \delta$ relationship between the input and output data. Coordinate backpropagation was formulated in a formal, abstract sense in terms of Lie Group actions on the metric fibre bundle. The error backpropagation algorithm was then formulated in terms of Lie group actions on the frame bundle. For a residual network in the limit, the Lie group acts smoothly along the fibres of the bundles. Experiments were conducted to confirm and better understand aspects of this formulation.

## 9 Acknowledgements

This work has been supported in part by the U.S. Air Force Office of Scientific Research (AFOSR) under Grant No. FA9550-15-1-0400. The first author has been supported by PSU/ARL Walker Fellowship. Any opinions, findings and conclusions or recommendations expressed in this publication are those of the authors and do not necessarily reflect the views of the sponsoring agencies.

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
