[Supplementary Material]

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

# 10 Appendix

This appendix includes the proof that the infinite product of matricies converges. Additionally in this appendix are the geometric definitions [18] necessary to formulate the principal and associated fibre bundles.

As $L \to \infty$, Equation 12 becomes an infinite product of matricies (from our infinite applications of the chain rule). Analogous to the scalar case, $P_{.a_0}^{a_L \cdot} = \prod_{l'=0}^{L-1} (\delta_{.a_{l'}}^{a_{l'+1}\cdot} + A_{.a_{l'}}^{a_{l'+1}\cdot})$ converges if $\sum_{l'=0}^{L-1} ||A_{.a_{l'}}^{a_{l'+1}\cdot}||_2$ converges [19]. For a fully connected network with activation $\tanh(z)$, the following inequality holds:

$$\sum_{l'=0}^{L-1} ||A_{.a_{l'}}^{a_{l'+1}\cdot}||_2 = \sum_{l'=0}^{L-1} || \left( \frac{\partial f(z^{(l'+1)}; l')}{\partial z^{(l'+1)}} \right)_{.e_{l'+1}}^{a_{l'+1}\cdot} \left( \frac{\partial z^{(l'+1)}}{\partial x^{(l')}} \right)_{.a_{l'}}^{e_{l'+1}\cdot} \Delta l ||_2 \leq$$

$$\sum_{l'=0}^{L-1} 2 \cdot ||W_{.a_{l'}}^{e_{l'+1}\cdot}||_2 \Delta l = 2 \cdot E \left[ ||W_{.a_{l'}}^{e_{l'+1}\cdot}||_2 \right] < \infty \quad (16)$$

where $||.||_2$ is the $\ell^2$ norm and $E[.]$ is the expectation. This shows that the infinite sum converges, implying that in the limit Equation 12 converges. In the limit the actions of the coordinate transformations on the metric tensor smoothly transform the metric tensor coordinate representation along the fibres of the associated bundle.

We will now define the structures required to abstractly formulate neural Lie Group actions on the principal and associated fibre bundles.

**Definition 10.1.** (Bundle) A bundle is a tuple $(E, \pi, B)$ in which $E$ and $B$ are topological manifolds and $\pi : E \to B$ is a surjective map. $E$ is called the entire space or total space, $B$ is called the base space and $\pi$ the projection map.

**Definition 10.2.** (Fibre bundle) A fibre bundle is a tuple $(E, \pi, B, F)$ in which $E$, $B$ and $F$ are topological manifolds and $\pi : E \to B$ is a surjective map and locally we have, for $U \subset B$, $U \times F = E$. $F$ is called the fibre over $B$.

**Definition 10.3.** (Section) A section on a fibre bundle $(E, \pi, B, F)$ is a map $\sigma : B \to E$ such that $\pi \circ \sigma = 1_B$, where $1_B$ is the identity map on $B$.

**Definition 10.4.** (Trivial fibre bundle) A trivial fibre bundle is a fibre bundle $(E, \pi, B, F)$ in which $E = B \times F$ everywhere.

*Remark.* (Fibre bundle) A helpful way to understand these definitions is with vector fields on the surface of a sphere $S^2$. A vector field on the surface of a sphere is a section of a fibre bundle, as at each basepoint $p \in B = S^2$ there exists a vector $v \in F = V$ such that $\sigma(p) = (p, v) \in U \times F \subseteq E$. A different section can give a different vector field. The surjective map $\pi : E \to B$ is then defined as $\pi(p, v) := p$, as projecting the pair down to the basepoint. If the section is a metric field satisfying the metric axioms, then there is a notion of distance, and thus shape, to the base space $B = S^2$.

**Definition 10.5.** (Lie Group) A Lie Group $(G, \bullet)$ is

(i.) a group with group operation $\bullet$

(ii.) $G$ is a smooth manifold and the maps $\mu$ and $\iota$ are smooth (in the topological sense).

(a.) $\mu : G \times G \to G$ where $\mu(h_1, h_2) := h_1 \bullet h_2$

(b.) $\iota : G \to G$ where $\iota(h) := h^{-1}$

**Definition 10.6.** (Lie Group left action) Let $(G, \bullet)$ be a Lie Group and $M$ a smooth manifold and the smooth map $\rhd : G \times M \to M$ satisfies

(i.) $e \rhd p = p \, \forall p \in M$ and $e \in G$ is the group identity element.

(ii.) $h_2 \rhd (h_1 \rhd p) = (h_2 \bullet h_1) \rhd p \, \forall p \in M$ and $\forall h_1, h_2 \in G$.

Any such smooth map $\rhd$ is called a left $G$-action on $M$.

**Definition 10.7.** (Lie Group right action) Let $(G, \bullet)$ be a Lie Group and $M$ a smooth manifold and the smooth map $\lhd : M \times G \to M$ satisfies

(i.) $p \triangleleft e = p \; \forall p \in M$ and $e \in G$ is the group identity element.

(ii.) $(p \triangleleft h_1) \triangleleft h_2 = p \triangleleft (h_1 \bullet h_2) \; \forall p \in M$ and $\forall h_1, h_2 \in G$.

Any such smooth map $\triangleleft$ is called a right $G$-action on $M$.

*Remark.* (Lie Group right action) Let $\triangleright \colon G \times M \to M$ be a left $G$-action, then define the right $G$-action $\triangleleft \colon M \times G \to M$ in terms of the left as $p \triangleleft h := h^{-1} \triangleright p$. The left $G$-action can be thought of as transforming the basis while the right $G$-action can be thought of as transforming the tensor components. A point $p$ has components $p^a$ in the basis $e_a$. When transforming the basis $e_a$ to $h^c_{.a} e_c$, one must also transform the components $p^a$ to $p^b (h^{-1})^a_{.b}$. Altogether, this ensures that the point does not depend on the choice of basis: $p = p^a e_a = p^b (h^{-1})^a_{.b} h^c_{.a} e_c = p^b \delta^c_{.b} e_c = p^b e_b = p$

**Definition 10.8.** (Lie Group properties) Let $\triangleright \colon G \times M \to M$ be a left action.

(1.) $\forall p \in M$, define its orbit under the action as $\mathcal{O}_p := \{ q \in M \mid \exists h \in G : h \triangleright p = q \}$.

(2.) $p$ and $q$ in $M$ are equivalent if they lie on the same orbit, i.e. $p \sim_G q : \iff \exists h \in G : q = h \triangleright p$

(2.1) This is an equivalence relation and can be used to construct the quotient set $M/G := M/\sim_G$

(3.) $\forall p \in M$ the stabilizer $S_p := \{ h \in G \mid h \triangleright p = p \}$

(3.1) The action $\triangleright$ is free if $\forall p \in M$ the stabilizer $S_p = \{ e \}$ where $e \in G$ is the identity element.

*Remark.* (Lie Group properties) It is helpful to think of the orbit $\mathcal{O}_p$ is the set of coordinate representations of the point $p \in M$ possible by the group $G$.