[Reviews · NeurIPS 2017]

Reviewer 1



This paper interprets the deep residual networks from an interesting perspective, i.e., based on the Riemannian geometry. First, the authors use dynamical systems to explain the performance of the residual network, compared with the ordinary static networks. Thus, given an initial condition/input data, the differentiable equation maps this point to its target. As the number of layer goes to infinity, the authors show that the metric tensor for residual networks converges and is smooth, and thus defines a Riemannian manifold. Experiments validate part of the proposed theory. There are two suggestions about the paper. 1. When illustrating (residual) deep neural network based on the Riemannian geometry, it is better to compare with the latent variable explanation. The author introduces these two perspectives in section 1, but, I think, there need more discussions and comparisons in the derivations of the theory and in the experiments. 2. In addition to the illustration, is there any insights for the design of deep neural network based on the Riemannian geometry?

Reviewer 2



This paper is an attempt at explaining the mechanism at work in deep learning methods. The authors propose to Interpret neural networks as methods for solving differential equations. So far, we could not deny the effectiveness of deep learning approaches but were eager to understand it. This paper seems to contribute to that line of research. Although my understanding of the technical arguments are limited, I believe that it could be of great interest to the NIPS community. I found the following papers : - Building Deep Networks on Grassmann Manifolds - Huang et al. - Generalized BackPropagation Etude De Cas : Orthogonality - Harandi et al. - A Riemannian Network for SPD Matrix Learning - Huang et al. -Geometric deep learning: going beyond Euclidean data - Bronstein et al. which integrate notions of Riemannian geometry into a deep learning approach. What are the links between the submitted paper and those four approaches ? Finally, could the authors makes some connections between their interpretation of deep learning and natural gradients ? After reading my fellow reviewers' comments and the authors' rebuttal, I am still convinced that the paper is a good fit for NIPS.

Reviewer 3



The paper develops a mathematical framework for working with neural network representations in the context of finite differences and differential geometry. In this framework, data points going though layers have fixed coordinates but space is smoothly curved with each layer. The paper presents a very interesting framework for working with neural network representations, especially in the case of residual networks. Unfortunately, taking the limit as the number of layers goes to infinity does not make practical application very easy and somewhat limits the impact of this paper. The paper is not always completely clear. Since the goal of the paper is to present a minority perspective, clarity should be paramount. experiments are a bit disapoiting. Their goal is not always very explicit. What the experiments do exactly is not very clear (despite their obvious simplicity). The experiments which show how a simple neural networks disentangle data points feel well known and their relation to the current paper feels a bit tenuous. Line 62: The . in tensors (…) ensure consistency in the order of the superscrips and subscripts. => Not very clear. I assume the indices with a dot at the end are to be on the left of indices with a dot at the beginning (so that the dots would sort of align). line 75, Eq 1: Taking the limit as L -> /infty seems to pose a problem for practical applications. Wouldn’t an equation of the form bellow make sense (interpolation between layers): x^a(x + \delta l) = x^a(l) + f^a(x^b(l); l) \delta l instead of x^a(x + 1 ) = x^a(l) + f^a(x^b(l); l) \delta l 6.1 & Figure1: The Harmonic oscillator Section 6.1 and the corresponding figures are unclear. => The given differential equation does not indicate what the parameters are and what the coordinates are. This is a bit confusing since \xi is a common notation for coordinates in differential geometry. We are told that the problem is two dimensional so I assume that x is the two dimensional variable. => Figure 1 is confusing. What does the finite differencing plot show ? what do the x and y axis represent ? What about metric transformations ? What about the scalar metric values ? The fact that the particle stands still is understandable if in the context of a network but confusing w.r.t. the given differential equation for x. => a, b = 1,2 is not clear in this context. Does this mean that a = 1 and b = 2 ? can the state space representation be written in the given way only if a, b = 1,2 ? Figure 2: Every plot has the legend “layer 11”. Typo ? section 6.3 is a bit superfluous.